

# Focus on Polish nurses' health condition: a cross-sectional study

Anna Bartosiewicz[1], Edyta Łuszczki[1], Pawel Jagielski[2], Lukasz Oleksy[3,4], Artur Stolarczyk[5] and Katarzyna Dereń[1]

[1] Institute of Health Sciences, Medical College of Rzeszow University, Rzeszów, Poland
[2] Department of Nutrition and Drug Research, Institute of Public Health, Faculty of Health Sciences, Jagiellonian University Medical College, Kraków, Poland
[3] Physiotherapy and Sports Centre, Rzeszow University of Technology, Rzeszów, Poland
[4] Oleksy Medical & Sports Sciences, Łańcut, Poland
[5] Orthopedic and Rehabilitation Department, Medical University of Warsaw, Warsaw, Poland

## ABSTRACT

**Background:** The nursing profession carries with it many negative factors and a high risk of developing chronic diseases, including overweight and obesity. According to statistics, the health condition of this professional group is much worse than that of the average population of the same age. As the largest and most trusted professional group in the world, nurses are critical to the health of any nation. The aim of the study was to assess the Polish nurse's health condition based on real measurements of parameters important for the occurrence of metabolic diseases.
**Methods:** This is the cross-sectional, conducted among two age groups of Polish nurses (<50 years and ≥50 years). Study included the measurements with DXA, the assessment of glucose concentration and lipid profile, the measurement of blood pressure and socio-demographic data of the surveyed nurses.
**Results:** The majority of respondents above 50 years old were nurses with elevated total cholesterol (79.3%), elevated LDL cholesterol (72.8%), 42.4% of studied nurses had hypertension.
**Conclusion:** Individual, local and national measures to prevent and support the health of this professional group are needed.

Corresponding author
Edyta Łuszczki, eluszczki@ur.edu.pl

## INTRODUCTION

Nurses are a key resource. They cover more than half of all health professionals in the world, providing essential services throughout the health care system (*World Health Organization, 2020*). The health and good condition of this professional group is a guarantee of the quality of care and safety of the patients (*Heath, 2019*; *Rosa et al., 2019*). However, the specificity of the work of nurses is daily exposure to many harmful factors that may adversely affect their health, significantly reduce the efficiency of their work, and contribute to an increase in the number of adverse events. The reasons for these risks may be work under time pressure, exposure to physical, chemical, and biological factors, shift work, staff shortages, work in conditions that endanger the patient's life, as

well as organizational and local conditions (*Rosa et al., 2019*; *Letvak, 2012*; *Siebenhüner, Battegay & Hämmig, 2020*; *Samur & Seren, 2019*; *Suliman & Aljezawi, 2018*).

Among the factors that pose a threat in the workplace of nurses, we can distinguish:

– biological, chemical and physical factors;
– factors related to working conditions;
– psychophysical factors (*Rosa et al., 2019*; *Letvak, 2012*; *Siebenhüner, Battegay & Hämmig, 2020*; *Samur & Seren, 2019*; *Suliman & Aljezawi, 2018*; *Kowalczuk, Krajewska-Kułak & Sobolewski, 2019*).

Biological agents are all factors causing infectious diseases (viruses, bacteria, fungi, protozoa), allergens (bacteria, fungi) and biological toxins (endotoxins, mycotoxins). They promote infections of the upper respiratory tract, conjunctiva, eyes, and skin, poisoning and allergies. A biological risk is also posed by single-use medical equipment (syringes and needles) commonly used by nurses (*Samur & Seren, 2019*; *Suliman & Aljezawi, 2018*). Special attention should be paid to viruses that cause hepatitis (HBV, HCV) and acquired immunodeficiency virus (HIV) (*Azizoğlu, Köse & Gül, 2019*; *Gajewska & Sienkiewicz, 2018*; *Marcinkowski, 2003*). The most common cause of accidents at work and infections of nurses are occupational exposures resulting from punctures, cuts, or contamination of the skin and mucous membranes with blood or other biological material from patients (*Samur & Seren, 2019*; *Suliman & Aljezawi, 2018*; *Azizoğlu, Köse & Gül, 2019*). According to the avaiable data, a nurse in his daily work meets 25 pathogens transmitted through the blood and mucous membranes (*Gajewska & Sienkiewicz, 2018*; *Marcinkowski, 2003*). Chemical factors present in the nurse's workplace include various types of disinfectants and medications, and personal protective equipment. They irritate the conjunctiva of the eyes, skin, and respiratory tract, which are often allergenic and sometimes even carcinogenic. Physical factors present in the nurse's workplace include noise, radiation, and work in an electromagnetic field (*Samur & Seren, 2019*; *Suliman & Aljezawi, 2018*; *Latina et al., 2020*; *Fagundo-Rivera et al., 2020*).

Factors resulting from working conditions, such as shift work, overload with duties and long working hours, pose a major threat to the health of nurses. According to the analyzed studies, disturbances in the rhythm of sleep and irregularity of consumed meals are a direct cause of problems with the digestive system, circulatory system, spine and even disorders of the menstrual cycle (*Kliszcz et al., 2004*; *Bardhan et al., 2019*). Research shows a strong correlation between shift work and an increased risk of obesity, which in turn leads to the development of many metabolic diseases. According to researchers the working environment of a nurse and the need of performing specific professional functions that predispose this professional group to greater susceptibility to the occurrence of civilization and occupational diseases (*Rosa et al., 2019*; *Gajewska & Sienkiewicz, 2018*; *Kułagowska & Kosińska, 2005*). The next group of threats are psychophysical factors, which include physical stress (static and dynamic) and neuromental stress (emotional stress, stress, depression, occupational burnout) (*Kowalczuk, Krajewska-Kułak &*

*Sobolewski, 2019; Bardhan et al., 2019; Kowalczuk, Krajewska-Kułak & Sobolewski, 2020; Kowalczuk, Krajewska-Kułak & Sobolewski, 2020*).

Working for many hours in a forced body position and excessive physical exertion contribute to degenerative changes in the muscular, skeletal, and joint systems as well as chronic back and spine pain (*Gajewska & Sienkiewicz, 2018*). These ailments are relatively common, they occur in the vast majority of nurses over 50 (90%), significantly reducing the effectiveness of their work (*Burdelak & Pepłońska, 2013*).

Shift work not only during the week, but also on Sundays and holidays, significantly disturbs the social and family functioning of the study group. Contact with accident victims and dealing with terminally ill patients contribute to the development of chronic stress and, consequently, to the development of occupational burnout. A common problem among nurses is depression, which affects almost two-thirds of this professional group of varying intensity (*Rosa et al., 2019; Books et al., 2017*).

This is due to the increasingly more frequent patient attitude and social misunderstanding. Researchers point out that hard work, stress, sleep and eating disorders lead to various health complications, in particular cardiovascular diseases, neurological disorders, and weaknesses of immunity (*Gajewska & Sienkiewicz, 2018; Sienkiewicz, Paszek & Wrońska, 2007; Gao et al., 2012*). This translates into the quality and length of their lives (*Ministry of Health, 2020*). The data in this respect is also disturbing. According to calculations from the last 5 years, the average age of nurses at the time of death is only 61.5 years, which is much lower than for the entire female population, which in Poland is 81.8 years (*Ministry of Health, 2020; Majewska, 2020; Zdrowia, 2021*). Moreover, Priano, conducting a literature review, reports that the health of as many as 60–74% of American nurses is endangered due to a lack of physical activity and in 53–61% due to bad eating habits (*Priano, Oi Saeng & Jyu-Lin, 2018*).

Due to their education, nurses are aware about the importance of health-promoting activities, such as healthy eating, physical activity, coping with stress, sleep hygiene, the ability to deal with stress and maintaining healthy relationships (*Mak et al., 2018*). However, work overload, the need to work in two or more jobs for financial reasons, and therefore lack of time may result in the recommended healthy lifestyle guidelines not being followed, and this knowledge may not translate into self-care of nurses (*Kang, 2020*). An example would be the study of the Cardiovascular Nurses Association, where most of nurses working in the field of prevention of heart disorders had a history of hypertension, lipid disorders, and obesity (*Fair, Gulanick & Braun, 2009*). Studies conducted among American women show that nurses have a lower level of participation in preventive examinations and a healthy lifestyle than the population of other women in this country, and nurses' knowledge in this area rarely translates into their self-care (*Chomistek et al., 2015*). Long working hours, overload of shift work and a stressful work environment mean that they do not find enough motivation and strength to take care of their health (*Ross et al., 2017; Tucker et al., 2010; Franek, Bartusek & Czajkowska, 2015*). The Women's Health Study conducted by Harvard Medical School and Brigham and Women's Hospital among nearly 40,000 healthcare professionals showed that nurses rarely cared about their health behavior, despite they are engaging in promoting the health of other women

(*Harvard Medical School, Brigham and Women's Hospital, 2004*). In addition, previous research shows that there is a lack of good practices among nurses regarding proper nutrition or regularity of meals (*Priano, Oi Saeng & Jyu-Lin, 2018*; *Nahm et al., 2012*).

In such a critical situation, it is important not only to care for the new medical staff, but also to care for the best health condition of currently working nurses (*Samur & Seren, 2019*). The need to stay healthy and to identify effective and cost-effective interventions of this occupational group is of particular importance in the face of an ongoing pandemic and aging nursing workforce in order to achieve better health behaviors and identify potential risk factor profiles (*Admi et al., 2008*; *Chidiebere, Tibaldi & La Torre, 2020*).

The justification for undertaking the research is the necessity to pay attention to such an important professional group to take preventive measures. Support and help in the treatment of existing diseases among nurses with longer work experience will allow for a better and longer development of the potential of educated medical staff. For young nurses starting their work, it will be an extreme fear to avoid neglect resulting in weakening their health condition. The nursing profession in most countries of the world, also in Poland, is performed mainly by women, and men constitute only a small percentage of this professional group. It is predisposed to a significant deterioration of health during the menopause. Therefore, it is important to pay attention to the health condition of nurses, especially that the vast majority of nurses in Poland and around the world are at this age (*World Health Organization, 2020*; *Zdrowia, 2021*). The aim of the study was to assess the Polish nurse's health condition based on real measurements of parameters important for the occurrence of metabolic diseases.

## MATERIALS AND METHODS

### Design and study participants

That was a cross-sectional, descriptive study, conducted in the 2020 among professionally active nurses in the south-eastern part of Poland (Subcarpathian district). Invitations to participate in the study were send *via* the website of the District Chamber of Nursing and *via* social media to 25 randomly selected medical entitles selected *via* randomized algorithm programme using the EPI INFO (StatCalc) software. Assuming the number of professional active nurses in the Podkarpackie Voivodeship, (aprox. 12,400), the sample size included 153 people and amounted to a confidence level of 95% and 8% margin of error. The information contained data on the planned measurements. Among the invited medical entities, six gave positive feedback. Information on planned measurements was sent to these centers again. The survey questionnaire, recommendations for participation in the study, proposed dates and times of measurements and contact to the research team for nurses willing to participate in the measurements were made available. Nurses interested in the study could call the indicated telephone number and set a convenient date for measurements as well as obtain answers on their questions.

The following criteria for the nurse's section was adopted:

– consent to participate in measurements
– professionally active nurses

– no symptoms of the disease within the last 2 weeks.

Nurses working at the institution that agreed to participate in the measurements were fully informed in writing and verbally about the nature of the study. They were assured of the voluntary, anonymous, and free of charge participation in the measurements. In order to ensure the confidentiality of data, each participant was assigned an identification number and the questionnaires were numbered. The encoded envelope attached to the questionnaire ensured an anonymous return of the data and its relation with the person's measurements. Finally, 156 nurses reported willingness to participate in the study, three of them were excluded because they did not meet the study inclusion criteria (pregnancy and infection). All measurements were carried out in the Natural and Medical Center for Innovative Research of the Rzeszow University, Rzeszów, Warzywna Street 1A (building G-4 and G-5) for a period of 3 weeks (from February 3 to 21st, 2020), in the morning between 9 am to 12 pm. Data from 153 subjects were included in the statistical analysis (Fig. 1).

Measurements included: blood pressure, fasting glucose, lipid profile and body mass composition. 156 nurses participated in the measurements, three of them were excluded because they did not meet the study inclusion criteria (pregnancy and infection). Data from 153 subjects were included in the statistical analysis. To check differences between the health condition of the surveyed nurses we divided the study participants into two age groups (<50 years and ≥50 years).

*Body Mineral Density cut point criteria:*
Normal: from +1.0 to −1.0
Osteopenia (early stage of osteoporosis): from −1.0 to −2.4
Osteoporosis: up to −2.5 and less
Advanced osteoporosis: −2.5 (*Kirschner et al., 2011*).
The study methodology has been published in detail (*Bartosiewicz et al., 2021*).

### Statistical analysis

The estimation method and the following statistical methods were used: in order to present the data, the method of descriptive statistics was used: and standard deviation (SD), a statistical measure of scattering the results around the expected value, number (N), percentage (%), mean ($\bar{x}$), median (Me). To check the normality of the data the Shapiro–Wilk test was perform. The differences between quantitative variables were tested using the non-parametric Mann-Whitney U test. The analysis of variables having the character of qualitative data was carried out with the Pearson chi-square test.

To determine the position of a given result against the results of the reference group or population, the division into quartiles was used, *i.e.*, four congregations with an equal number of people:

– first quartile (Q1)-25% of observations are below and 75% above

– the second quartile (Q2) (*i.e.*, the median divides the set of observations into two equal parts

– third quartile (Q3)-75% of observations are below and 25% above.

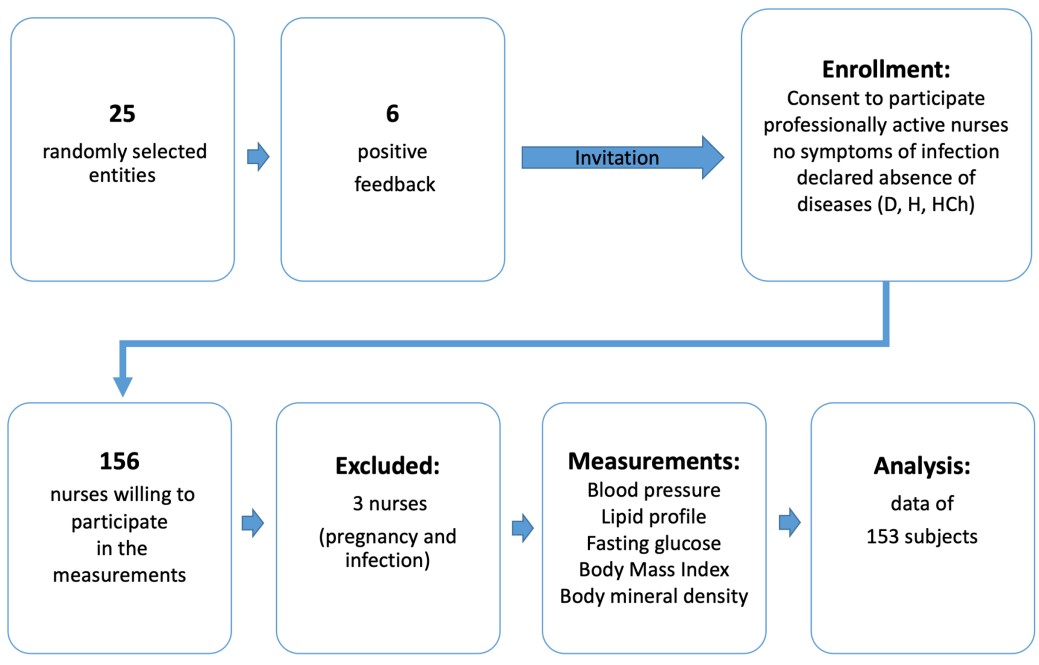

**Figure 1** **Flow chart demonstrating the selection of the study participants.** Explanation of abbreviations: D, diabetes; H, hypertension; HCh, hypercholesterolemia.

The statistical analyses were performed using PS MAGO PRO 6.0 (IBM SPSS STATISTICS 26).

Statistical significance was set at $p < 0.05$.

The power of the test was calculated using STATISTICA 13.3 for the results relating to the percentage of women with total cholesterol levels above normal. After the introduction Data on the percentage of people with elevated cholesterol and the size of the groups were introduced into the program. Additionally, as a gold standard two tailed test was considered and probability of Type I Error at a level 0.05 was assumed. Under the above constraints, we got highly satisfactory power of test equal to 0.8.

### Ethics

This research project was carried out in accordance with the Helsinki Declaration. The study was approved by the institutional Bioethics Committee at the University of Rzeszów (Resolution No. 30/05/2019) and all appropriate administrative bodies. Both the guardians and the participants gave their informed written consent to participate in the study.

## RESULTS

A total of 153 nurses aged 24 to 64 were examined. All of the study participants were women. The mean age of the respondents was 48.88 ± 8.57 years. 39.9% from study participants were women aged below 50 years (39.9%) and above 50 years (60.1%). There was a significant difference ($p < 0.0001$) in the education between the study participants. Higher education had younger nurses. In addition, a significant difference was in

**Table 1 Socio-demographic characteristic of the study participants.**

| Characteristics | Total (*n* = 153)% | <50 years (*n* = 61)% | ≥50 years(*n* = 92)% | *p*-value |
|---|---|---|---|---|
| **Place of residence** | | | | |
| City | 83.0 | 90.2 | 78.3 | 0.0549 |
| Village | 17.0 | 9.8 | 21.7 | |
| **Workplace** | | | | |
| Hospital | 47.1 | 54.1 | 42.4 | 0.1555 |
| Ambulatory | 52.9 | 45.9 | 57.6 | |
| **Work system** | | | | |
| One shift work | 51.0 | 49.2 | 52.2 | 0.7168 |
| Shift work and night duty | 49.0 | 50.8 | 47.8 | |
| **Education** | | | | |
| Basic nursing education | 33.3 | 11.5 | 47.8 | **<0.0001** |
| Bachelor of nursing | 20.3 | 23.0 | 18.5 | |
| Nursing (master's degree) | 46.4 | 65.6 | 33.7 | |
| **Additional qualifications** | | | | |
| No | 20.3 | 19.7 | 20.7 | 0.8826 |
| Yes | 79.7 | 80.3 | 79.3 | |
| **Self-assessment of health condition** | | | | |
| Very good | 14.4 | 18.0 | 12.0 | 0.5139 |
| Good | 64.7 | 60.7 | 67.4 | |
| I have no opinion | 15.7 | 18.0 | 14.1 | |
| Bad | 5.2 | 3.3 | 6.5 | |
| **Participation in preventive examinations** | | | | |
| No | 65.4 | 83.6 | 53.3 | **<0.0001** |
| Yes | 34.6 | 16.4 | 46.7 | |

Notes:
*p* - *p*-value, indicate significant values (*p* < 0.05).
Bold values denote statistical significance at the *p* < 0.05 level.

participation in preventive examinations. Older women participated in examinations significantly more often (*p* < 0.0001). Descriptive characteristics of the study group are presented in Table 1.

The values of individual parameters in the total of 153 nurses are presented in the Table 2. There were significant differences between two groups in most of the variables. Among older nurses a significantly higher level of triglycerides, LDL cholesterol, glucose, SBP, DBP and fat mass were observed. There was also a lower level of T-score BMD in women above 50 years old. On the other hand, younger nurses had higher level of HDL cholesterol.

In the Table 3 variables have been presented as a category. We found a significant differences between the younger and older nurses (*p* < 0.05) in total cholesterol, HDL cholesterol, LDL cholesterol, triglycerides, glucose and blood pressure. The majority of respondents above 50 years old were nurses with elevated total cholesterol (79.3%), elevated LDL cholesterol (72.8%). In addition, 48.9% of older nurses had elevated

Table 2 Values of individual parameters in the study group.

| Variable | Total (n = 153) | | | <50 years (n = 61) | | | ≥50 years (n = 92) | | | p-value |
|---|---|---|---|---|---|---|---|---|---|---|
| | Me | Q25 | Q75 | Me | Q25 | Q75 | Me | Q25 | Q75 | |
| Age (years) | 51.0 | 47.0 | 55.0 | 45.0 | 36.0 | 47.0 | 54.0 | 52.0 | 57.0 | **<0.0001** |
| TC (mg/dl) | 214.0 | 187.0 | 241.0 | 195.0 | 170.0 | 227.0 | 229.5 | 202.0 | 253.0 | 0.0697 |
| HDL (mg/dl) | 56.0 | 49.0 | 66.0 | 59.0 | 50.0 | 69.0 | 56.0 | 46.0 | 65.0 | **0.0001** |
| Triglycerides (mg/dl) | 127.0 | 90.0 | 180.0 | 102.0 | 81.0 | 130.0 | 148.0 | 105.0 | 206.5 | **<0.0001** |
| LDL (mg/dl) | 128.0 | 104.0 | 150.0 | 116.0 | 90.0 | 132.0 | 136.0 | 113.0 | 169.5 | **0.0001** |
| Glucose (mg/dl) | 93.0 | 87.0 | 101.0 | 90.0 | 85.0 | 94.0 | 96.0 | 89.5 | 104.0 | **0.0001** |
| SBP (mmHg) | 125.0 | 114.0 | 142.0 | 120.0 | 109.0 | 132.0 | 128.0 | 120.5 | 148.0 | **0.0012** |
| DBP (mmHg) | 78.0 | 71.0 | 86.0 | 74.0 | 68.0 | 80.0 | 80.0 | 74.0 | 87.5 | **<0.0001** |
| BMI (kg/m$^2$) | 26.0 | 21.9 | 29.7 | 23.1 | 20.4 | 26.6 | 27.4 | 24.4 | 30.7 | 0.1317 |
| BMD (g/cm$^2$) | 1.2 | 1.1 | 1.3 | 1.2 | 1.1 | 1.3 | 1.2 | 1.1 | 1.2 | **0.0035** |
| T- SCORE BDM | 1.0 | 0.2 | 1.7 | 1.4 | 0.6 | 2.0 | 0.8 | −0.2 | 1.5 | **0.0088** |
| FM (%) | 37.1 | 31.3 | 41.5 | 33.7 | 28.5 | 39.1 | 38.6 | 34.6 | 44.0 | **<0.0001** |

Notes:
Q25, lower quartile; Q75, upper quartile; FM, fat mass; BMI, Body mass index; TC, Total cholesterol; HDL, high-density lipoprotein; LDL, low-density lipoprotein; SBP, systolic blood pressure; DBP, diastolic blood pressure; Me, median; BMD, body mineral density; T-score BMD, osteoporotic fracture risk assessment, the result of comparing the patient's BMD with the density of the "young adult" at the age when the bones are strongest.
Bold values denote statistical significance at the $p < 0.05$ level.

triglycerides, compared to 19.7% of younger nurses. In the older group, 42.4% of women had hypertension and only 18% in younger group (Table 3).

## DISCUSSION

The aim of the study was to assess the Polish nurse's health condition according to age groups based on real measurements of parameters important for the occurrence of metabolic diseases. 153 nurses aged 24 to 64 participated in our study. We examined both the basic parameters and advanced measurements indicating the actual health condition of the nurses under study and the risk of developing metabolic diseases. Nurses participating in the study assessed their health as good (64.7%), however, it is worrying that more than half of the respondents (65.4%) declare that they did not participate in preventive examinations. It seems obvious that medical education translates into high awareness and compliance with preventive recommendations. The obtained results showed that there is a significant difference between the health condition of nurses over 50 years old compared to their younger colleagues. While these results seem to be obvious, they show how important it is to pay attention and take preventive action to avoid the risk of developing many diseases over the years. Earlier studies show that older age significantly reduces the health condition of nurses (*das Merces et al., 2019*). In a study by Conceicao at al. even though more than half (52.2%) of the nurses participating in the survey were under 35 years of age, as many as 24.4% had factors confirming the risk of developing metabolic diseases (*das Merces et al., 2019*; *Vgontzas et al., 2000*). The issue of the proper health condition of nurses is also important due to the shortage of nurses all

**Table 3 Category of variables in the study group.**

| Variable | Total (n = 153)% | <50 years (n = 61)% | ≥50 years (n = 92)% | p-value |
|---|---|---|---|---|
| **Total cholesterol** | | | | |
| Normal (150–190 mg/dl) | 28.8 | 41.0 | 20.7 | 0.0065 |
| Elevated (>190 mg/dl) | 71.2 | 59.0 | 79.3 | |
| **HDL** | | | | |
| Normal (>40 mg/dl) | 90.8 | 96.7 | 87.0 | 0.0403 |
| Below standard (<40 mg/dl) | 9.2 | 3.3 | 13.0 | |
| **LDL** | | | | |
| Normal (<115 mg/dl) | 35.9 | 49.2 | 27.2 | 0.0055 |
| Elevated (>115 mg/dl) | 64.1 | 50.8 | 72.8 | |
| **Triglycerides** | | | | |
| Normal (35–150 mg/dl) | 62.7 | 80.3 | 51.1 | 0.0002 |
| Elevated (>150 mg/dl) | 37.3 | 19.7 | 48.9 | |
| **Glucose** | | | | |
| Normal (70 to 99 mg/dL) | 71.2 | 86.9 | 60.9 | 0.0019 |
| Pre-diabetes (100 to 125 mg/dL) | 26.8 | 13.1 | 35.9 | |
| Diabetes (≥126 mg/dL) | 2.0 | 0.0 | 3.3 | |
| **Blood pressure** | | | | |
| Normal (120–129 mm Hg/80–84 mm Hg) | 59.5 | 70.5 | 52.2 | 0.0054 |
| Elevated (130–139 mm Hg/85–89 mm Hg) | 7.8 | 11.5 | 5.4 | |
| Hypertension (>140 mm Hg/90 mm Hg) | 32.7 | 18.0 | 42.4 | |

**Note:**
HDL, high-density lipoprotein, LDL, low-density lipoprotein.

over the world and the huge demand for nursing services due to aging societies (*World Health Organization, 2020*; *Drennan & Ross, 2019*). In many countries, the statistical age of a nurse is 52–53 years, which is the time when many diseases are physiologically worse. Reports show little interest of young people in the nursing profession, therefore, also in the context of staff shortages, proper prevention and care of this professional group is extremely important (*Ministry of Health, 2020*).

Our results show the health condition of Polish nurses, the results concerning nurses in the 50+ group are worrying. Analyzing the measurements, we see a significant, sometimes several times increase in the value of individual parameters in the group of older nurses. The majority of respondents above 50 years old were nurses with elevated total cholesterol (79.3%), elevated LDL cholesterol (72.8%). In the older group, 42.4% of women had hypertension. The BMI in the group of nurses studied slightly exceeds the norm (26.0), but when we compare the two analyzed age groups, the differences are significant (23.1 *vs* 27.4). The data of the report we refer to in the introduction indicate that the life expectancy of nurses is considerably shorter than that of statistical women, which also indicates a much worse health condition of this professional group (*Ministry of Health, 2020*; *Majewska, 2020*; *Zdrowia, 2021*). In the study *Woynarowska-Sołdan et al. (2018)* 44% of the surveyed nurses were overweight or obese, and even though the BMI in the whole

group was almost within the normal range (25.1), the BMI value increased with the age of the respondents. A study from Scotland found that seven out of 10 nurses in Scotland are overweight or obese, and the prevalence of overweight and obesity in Scotland is statistically significantly higher among nurses than other healthcare professionals and non-health workers. The authors of the study point to the urgent need for action to reduce the prevalence of overweight and obesity among the nursing staff in this country (*Kyle, Neall & Atherton, 2016*). This confirms previous international research that found higher prevalence of overweight and obesity among nurses than the general population in the UK, Australia, and New Zealand (*Bogossian et al., 2012*). The obtained results are also disturbing due to the fact that the nurses participating in the measurements were not aware of the actual parameters indicating their health condition. At the time of the measurements, most of them realized that they had high cholesterol values, too high glucose levels or hypertension. In a study by Miller et al. although the vast majority of nurses (96%) knew about the risk factors for developing heart disease, 26% were not aware of their diabetes, and about 90% were also unaware of hyperlipidemia (*Miller, Alpert & Cross, 2008*). Equally, the results of the Cardiovascular Nurses Association showed that 20%, 23% and 17% of nurses working in the field of heart disease prevention had a history of hypertension, lipid disorders and obesity (*Fair, Gulanick & Braun, 2009*). Faced with the poor health condition of health workers, including nurses, many countries are taking initiatives to support and promote a healthy lifestyle in this professional group.

An example is the huge undertaking and the Healthy Nurse, Healthy Nation-Grand Challenge campaign launched by the *American Nurse Association Enterprise (2017)*. It is an initiative that engages nurses and health care organizations to take action to improve their physical activity, sleep, nutrition, quality of life and safety for nurses. For those joining the program, the association provides a helpful online platform that inspires action, motivates, promotes a healthy lifestyle, collects data and provides support in undertaken activities. According to American research, nurses are less healthy than the average citizen, are more likely to be overweight, obese, have higher levels of stress, and shift the natural rhythm of sleep and social functioning through shift work. Nurses, as the largest and most trusted professional group in the world, are critical to the health of any nation. By taking care of their health, they can be good models for their patients, families, and friends. This is in line with Forencja Nightingale's recommendation that the nurse should be a good exemplar to others (*Dunphy, 2005*).

## Limitations and future research

To our knowledge, it is one of the first study in Poland where actual measurements of individual parameters were carried out and not based on the respondents' declarations. There are also several potential limitations of the study that need to be considered when interpreting the results. This study was limited in geographic scope and should be repeated among a larger sample and in more regions. Being the study cross-sectional, the causality and temporality issues should not be considered. More research is needed in the larger population of all age groups.

## CONCLUSIONS

Our study shows that the factors potentially considered risk factors for the development of metabolic diseases were more common among older nurses. Our study shows very well the possible consequences that may also occur among younger nurses in the future, if this professional group is not given special protection. High levels of overweight and obesity, hypertension and hypercholesterolemia are potentially harmful to the health of nurses, significantly limiting their efficiency also in the professional aspect. And although nurses have sufficient knowledge in the field of prevention and health promotion, there is often a discrepancy in the implementation of these activities in relation to each other. Research shows that nurses who are involved in taking care of their own health more often pass on to patients the benefits of making correct choices that promote a healthy lifestyle. Therefore, interventions at individual, local and national level are urgently needed. Nurses who already have risk factors for the development of metabolic diseases should be taken care of properly, young staff should be supported and motivated to take care of their health. Interventions among nursing students will help to educate healthy and conscious staff. The poor health condition of nurses means limited opportunities to educate and promote a healthy lifestyle among patients.

### Funding
The authors received no funding for this work.

### Competing Interests
Łukasz Oleksy is employed by Oleksy Medical & Sports Sciences (Orthoppeadic rehabilitation clinic).

### Author Contributions
- Anna Bartosiewicz conceived and designed the experiments, performed the experiments, authored or reviewed drafts of the paper, and approved the final draft.
- Edyta Łuszczki conceived and designed the experiments, performed the experiments, authored or reviewed drafts of the paper, and approved the final draft.
- Pawel Jagielski analyzed the data, prepared figures and/or tables, and approved the final draft.
- Lukasz Oleksy performed the experiments, analyzed the data, prepared figures and/or tables, and approved the final draft.
- Artur Stolarczyk analyzed the data, prepared figures and/or tables, and approved the final draft.
- Katarzyna Dereń performed the experiments, authored or reviewed drafts of the paper, and approved the final draft.

## Human Ethics

The following information was supplied relating to ethical approvals (*i.e.*, approving body and any reference numbers):

Bioethics Committee at the University of Rzeszów approved the study (Resolution No. 30/05/2019).

## Data Availability

The raw measurements are available in the Supplemental File.

## Supplemental Information

Supplemental information for this article can be found online at http://dx.doi.org/10.7717/peerj.13065#supplemental-information.

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
