# Peer review of "Focus on Polish nurses’ health condition: a cross-sectional study"

_PeerJ, doi:10.7717/peerj.13065_

## Round 0.1 · original submission · Minor Revisions

Thank you for your submission investigating the occurrence of metabolic disease in Polish nurses. Your manuscript was reviewed by 2 reviewers that both agree to minor revision.

Please address each point provided by the reviewers, particularly the following;

-Revise the introduction explaining the health problems related to occupational working conditions, which will allow for improved transition between sentences.

-Revise Table 3, and reword result outcome as suggested.

-Furthermore, the start of the Discussion highlights lifestyle perpectives of nurses, however, this is not the main aim or research question; therefore please revise accordingly.

Reviewer 1 ·

Basic reporting

In the introduction, many health problems experienced by nurses and related factors (time, physical, chemical, biological, infectious, etc.) are mentioned. However, it is necessary to create a transition between paragraphs. First of all, explaining the health problems related to occupational working conditions (such as shift work, long work and intensive work), then physical, chemical and biological etc. Indication of health problems that arise due to reasons can provide an integrity. Thus, it is thought that a better transition between sentences can be created.


What do the abbreviations D, H, HCh in Figure 1 mean? "Welling"?
Line 150. Typo. Parenthesis is not closed.
In order to accurately diagnose the health status of nurses; For each measurement parameter (blood pressure, height measurement, BMD etc.) and laboratory tests, it is recommended to indicate internationally valid reference ranges/cut-off points, with a bibliography.
Discussion part started with lifestyle perpectives of nurses. However, there is no such "subject" in the research questions or findings. It is recommended to write an introduction directly related to the findings. Otherwise, it is recommended to start the discussion on Line 293.
There is repeated informations about the findings of the research in the discussion. (lines 306-315). A specific discussion of the research findings is suggested.
Line 285-293: It can be moved to ıntro. It loks like associated with the importance of the subject.

Experimental design

In the method, the power of the sample to represent the universe should be explained by specifying the power analysis and accessibility ratio.

Validity of the findings

'no comment'

Reviewer 2 ·

Basic reporting

The research project is interesting and deals with important issues for the nursing profession. The work has a chance to be published, however, it requires improvement.

Experimental design

Table 3 requires a bit more description in the text of the work.

Validity of the findings

Only 153 nurses were included in the study. Therefore, it cannot be concluded that "... study shows the poor health condition of Polish nurses, especially those over 50", because the obtained results concern only a selected group of Polish nurses.

Additional comments

I propose to update the literature. Perhaps it is worth replacing older literature with more recent ones, especially on the health condition of Polish nurses. I present several possibilities:


The effect of subjective perception of work in relation to occupational and demographic factors on the mental health of Polish nurses. Frontiers in Psychiatry.

Working excessively and burnout among nurses in the context of sick leaves. Frontiers in Psychology.

Factors determining work arduousness levels among nurses: using the example of surgical, medical treatment, and emergency wards. BioMed Research International

---

## Round 0.2 · accepted · Accept

Dear authors,
Thank you for making the suggested edits, your manuscript has been approved for publication.